# Immune Checkpoint Blockade: A Strategy to Unleash the Potential of Natural Killer Cells in the Anti-Cancer Therapy

**DOI:** 10.3390/cancers14205046

**Published:** 2022-10-14

**Authors:** Melania Grottoli, Paolo Carrega, Lodovica Zullo, Chiara Dellepiane, Giovanni Rossi, Francesca Parisi, Giulia Barletta, Linda Zinoli, Simona Coco, Angela Alama, Silvia Marconi, Monica Parodi, Paola Orecchia, Sara Bassi, Massimo Vitale, Maria Cristina Mingari, Ulrich Pfeffer, Carlo Genova, Gabriella Pietra

**Affiliations:** 1UO Immunologia IRCCS Ospedale Policlinico San Martino, 16132 Genova, Italy; 2Laboratory of Immunology and Biotherapy, Department of Human Pathology, University of Messina, 98122 Messina, Italy; 3UO Oncologia Medica 2 IRCCS Ospedale Policlinico San Martino, 16132 Genova, Italy; 4DiMI, Department of Internal Medicine and Medical Specialties, University of Genova, 16132 Genova, Italy; 5UOS Tumori Polmonari IRCCS Ospedale Policlinico San Martino, 16132 Genova, Italy; 6DiMES, Department of Experimental Medicine, University of Genova, 16132 Genova, Italy; 7Laboratory of Tumor Epigenetics IRCCS Ospedale Policlinico San Martino, 16132 Genova, Italy; 8UO Clinica di Oncologia Medica IRCCS Ospedale Policlinico San Martino, 16132 Genova, Italy

**Keywords:** NK cells, immuno-checkpoint inhibitors (ICI), non-small cell lung cancer (NSCLC), PD-1

## Abstract

**Simple Summary:**

Immune checkpoint blockade (ICB) with antibodies targeting CTLA-4 (Cytotoxic Lymphocyte Antigen 4) and/or programmed death-1 protein (PD-1)/programmed death ligand-1 (PD-L1) has significantly modified the therapeutic landscape of a broad range of human tumor types, including advanced non-small-cell lung cancer (NSCLC). Despite great advances of checkpoint immunotherapies, a minority of NSCLC patients (<20%) respond and/or experience long-term clinical benefits from these treatments. Limited response rates of T cell–based checkpoint immunotherapies suggest the presence of other checkpoints able to inhibit effective anti-tumor immune responses. Natural Killer (NK) cells represent a promising target for tumor immunotherapies, particularly against tumors that escape T-cell-mediated control. Like T cell function, NK cell function is also regulated by inhibitory immune-checkpoint molecules. In this review, we will provide an overview of the rationale, mechanisms of action, and clinical efficacy of these NK cell-based checkpoint therapy approaches. Finally, the future directions and current enhancements planned will be discussed.

**Abstract:**

Immune checkpoint inhibitors (ICIs) immunotherapy has represented a breakthrough in cancer treatment. Clinical use of ICIs has shown an acceptable safety profile and promising antitumor activity. Nevertheless, some patients do not obtain clinical benefits after ICIs therapy. In order to improve and cure an increasing number of patients, the field has moved toward the discovery of new ICIs expressed by cells of innate immunity with an elevated inherent antitumor activity, such as natural killer cells. This review will focus on the recent findings concerning the role of classical and non-classical immune checkpoint molecules and receptors that regulate natural killer cell function, as potential targets, and their future clinical application.

## 1. Introduction

To date, cancer immunotherapy with immune checkpoint inhibitors (ICIs) has been a cornerstone of the treatment for several solid and hematologic malignancies; among these, non-small cell lung cancer (NSCLC) treatment has gained significant benefits from the introduction of ICIs, as such agents have been approved for use at virtually any disease stage, from resectable to advanced disease, both as a single-agent approach and as part of combination regimens with chemotherapy. Notably, the currently approved agents are directed on programmed death 1 (PD-1) protein or its ligand (PD-L1), as well as cytotoxic lymphocyte antigen 4 (CTLA-4), which mostly involve T cell lymphocytes. With reference to advanced NSCLC, single-agent PD-1 is the current standard of care for non-oncogene addicted NSCLC with high PD-L1 expression, while combinations involving ICIs and chemotherapy are employed with absent or low PD-L1 expression [1]. While the contribution of ICIs to NSCLC management is extremely relevant, there is still space for further development; indeed, it is known that the complexity of the immune system allows to consider other mechanisms. On one hand, the effect of PD-1/PD-L1 or CTLA-4 blockade on the antineoplastic activity of other immune cells, such as Natural Killer (NK) lymphocytes, still needs to be elucidated. Along this line, additional immune checkpoints (ICs) can be exploited with therapeutic intent, including specific NK-related ICs (including lirilumab and monalizumab), as well as other novel therapeutic immune targets. From a clinical perspective, the development of novel ICI-based regimens [2] might lead to the improvement of current practice; indeed, a current priority in the management of non-oncogene-addicted NSCLC is represented by the ability to overcome the resistance to established ICIs, which in turn is expected to result in improved response and survival [3]. This review aims to summarize the current state of knowledge of the effects elicited by ICIs on NK cells, as well as the novel immunotherapeutic agents likely to be soon introduced in clinical practice.

## 2. NK Cell Subsets Diversity

Traditionally, two main circulating NK cell populations have been defined based on their differential expression of CD56 and CD16 markers, termed CD56^bright^ and CD56^dim^ NK cells, which do not overlap for a number of phenotypic and functional properties [4,5]. These include the expression of a high level of perforins and granzymes, as well as, MHC class I-specific inhibitory Killer Ig-like Receptors (KIRs), which are restricted to CD56^dim^ NK cells and license this subset to mediate strong cytotoxic response upon the engagement of activating receptors (i.e., NCR, NKG2D, and DNAM-1). Moreover, the high expression of the Fcγ receptor CD16 provides CD56dim NK cells with the capacity to exert antibody-dependent cell-mediated cytotoxicity (ADCC). Conversely, CD56^bright^ NK cells are considered efficient cytokine producers endowed with immunoregulatory properties, but poorly cytotoxic, unless appropriately activated. A recent series of observations have greatly expanded on NK cell lineage diversity by showing that circulating NK cell subsets actually represent only a minor part of total NK cells in our body, as peripheral tissues harbor a relevant amount of “unconventional” subsets of NK cells that apparently do not recirculate in the blood or lymphatics, and possess distinct phenotypic profiles [6,7]. Tissue residency has now been described as a feature for several subsets of lymphocytes, such as NK cells, “helper”-like innate lymphoid cells (ILCs), and T cell subsets [8]. As for other lymphocytes, NK cells that reside in tissues display markers, such as CD69, CD103 (Integrin alpha E), CD49a (i.e., the α1 subunit of α1β1 integrin), and CXCR6, which are all involved in their retention within tissues and, practically, allow for their identification and isolation [9]. To date, discrete subsets of TR-NK cells have been identified in normal human districts such as the uterus (both at steady-state and during pregnancy), bone marrow (BM), secondary lymphoid organs (SLO), liver, and lung [6,7]. Recent advances have characterized several novel human (h)-NK subsets. Among them, adaptive NK cells demonstrate an intriguing, specialized antibody (Ab)-dependent response and several adaptive immune features [10]. Most adaptive NK cells express a high level of NKG2C and lack NKG2A, express KIRs, Leukocyte Ig-like Receptor B1 (LILR-B1/ILT2), low levels of natural cytotoxic receptors (NCRs) (i.e., NKp46 and NKp30), CD161, and T-cell immunoglobulin and mucin-containing domain (TIM-3). In addition to peripheral blood (PB) NK cells, adaptive NK cells can be detected the lymph nodes (LNs) tonsils, liver, pleural fluid, and other sites. Adaptive NK cells are often associated with prior human Cytomegalovirus (HCMV) infection [11]. Interestingly, adaptive NK cells can also infiltrate into tumor tissues, such as NSCLC [12]. Functionally they are able to mount ADCC, thus adaptive NK cells may play a part in tumor eradication by specific targeted Ab-based cancer therapy. In the tumor microenvironment (TME), a large number of immune-suppressive pathways are combined to inhibit NK cell function. In addition, the continuous exposure to tumor cells and the microenvironment contributes to the exhaustion of the immune effector cells. Exhausted Tumor-associated (TA)-NK cells exhibit downregulation of effector cytokines, decreased degranulation potential, downregulation of activating receptors (such as NKG2D), upregulation of inhibitory receptors such as PD-1, TIM-3, T cell immunoreceptor with Ig, and Immunoreceptor Tyrosine-based inhibition Motif (ITIM) domains (TIGIT) and NKG2A and decreased expression of Eomesodermin and T-bet transcription factors (TFs) [13,14]. Expression of the latter markers on NK cells correlates with decreased NK cell functionality and blockade of these receptors can increase NK cell cytotoxicity and function [14].

## 3. Expression of MHC Class I- Specific Inhibitory Receptors by NK Cells (Classical ICs)

NK cell functions depend on the balance between inhibitory and activating signals mediated by cell surface receptors.

Classical inhibitory receptors expressed by NK cells are mainly human leukocyte antigen (HLA) class I- binding molecules and include KIRs, CD94/NKG2A, and LILR-B1/ILT2. [15]. The family of inhibitory KIRs (iKIRs) includes the two domain KIR2DL1 and KIR2DL2/3 molecules and the three domain KIR3DL1 and KIR3DL2 receptors that recognize epitopes shared by different groups of classical HLA class I molecules. CD94/NKG2A is specific for the non-classical HLA-E molecules in complex with several peptides derived from the leader sequence of HLA-A, -B, or -C molecules [16,17] or from CMV [18,19]. LILR-B1/ILT2 displays a promiscuous recognition of many classical and non-classical HLA class I molecules as well as of the CMV-derived class I-like molecules UL18 [20].

## 4. Expression of Non-Classical ICs by NK Cells

NK cells also express other immunoregulatory receptors that function as non-classical IC. These include PD-1, TIGIT, Tactile/CD96, TIM-3, CD161, and Lymphocyte-activation gene 3 (LAG3) [21]. Resting NK cells usually do not express these molecules, while they can be often found on “stressed”/activated NK cells during infection or tumors.

PD-1 is an inhibitory receptor of cellular immune response expressed in different immune cell populations. PD-1 can bind some specific ligands, i.e., PD-L1 (B7-H1, CD274), or PD-L2 (B7-DC, CD273) [22].

These inhibitory receptors are required for peripheral tolerance generation and for the inhibition of damage during inflammation in peripheral tissues. While PD-L2 expression is mostly limited to dendritic cells (DC), macrophages, and lung cells, PD-L1 is most widely expressed and can be overexpressed on several tumor cells [23,24] thus, probably, favoring escaping of tumor cells from immune surveillance. PD-1 expression on NK cells was shown in patients with ovarian carcinoma, Kaposi sarcoma myeloma, and gastrointestinal cancers [25,26,27]. It is conceivable that the TME, through signals delivered by soluble factors and/or cells, can induce PD-1 expression [28]. Moreover, trogocytosis can act as a mechanism by which PD-1 is transferred from tumor cells to NK and T cells. PD-1 trogocytosis sharply inhibits the potential of NK cells to eradicate tumors in vivo [29]. Recently, it has been also reported that PD-1 is stored in cytotoxic granules and its surface expression increased following recognition of tumor cells, concurrent with CD107a surface mobilization [30] on NK cells. TIGIT is an inhibitory IC expressed by resting T and NK cells, upregulated along with activation [31] and often overexpressed in a variety of cancers [32]. TIGIT competes with CD226 to bind to PVR (CD155) and Nectin-2 (CD112) two molecules often up-regulated on tumor cells thus playing an important role in the NK-cell activity inhibition in part counterbalanced by the co-stimulatory activity exert by DNAM-1(CD226) receptor [33,34]. Tactile/CD96 is another important IC for NK cell effector functions, which shares with TIGIT the ligands. CD96^+^ NK cells are significantly increased in the intra-tumor tissues of hepatocellular carcinoma (HCC) [35,36]. CD96^+^ NK cells are functionally impaired with reduced capacity to release IFN-γ and TNF-α, and low gene expression levels of *Tbx21*, *Prf1*, and *GzmB*. TIM-3 is expressed in both adaptive and innate immune cells, but human NK cells transcribe the highest amounts of TIM-3 among lymphocytes [37].

So far, galectin-9, phosphatidylserine, and High Mobility Group Box (HMGB)1 have been described as TIM-3 ligands. To date, many aspects of the biology of TIM-3 have not yet been completely elucidated. Thus, while it has been reported that, to exert its function, TIM-3 should interact with Carcinoembryonic antigen related cell adhesion molecule (CEACAM)-1 (in cis and/or trans) [38], a recent study does not provide any evidence for an interaction between these receptors, suggesting that the inhibitory signaling in effector cells is mediated by TIM-3 cytoplasmic sequences [39]. LAG-3 is another inhibitory IC that can be found on activated NK cells [40]. In T cells, LAG-3 co-localizes with CD4 molecules in endosomes, secretory lysosomes, and MicroTubule Organizing center (MTOC) [41]. LAG-3 main ligands are MHC class II molecules, and the fibrinogen family protein 1 (FGL1) [42]. In humans, FGL1 is usually overexpressed by several tumors. In melanoma and lung cancer patients, elevated FGL1 plasma levels are correlated with resistance to immunotherapy with anti-PD-1 mAbs and poor outcome. These data indicate that the FGL1/LAG-3 interaction may aid tumor immune escape. Along this line, evidence that a combined blockade of FGL1 and PD-1 has a synergistic effect has been obtained in animal models [42]. Finally, CD161, belonging to the C-type lectin superfamily, is an inhibitory receptor that recognize Lectin-Like Transcript 1(LLT1), a ligand expressed by several tumors such as non-Hodgkin’s lymphoma (NHL) [43].

## 5. NK Cell Targeting to Improve Anti-Tumor Response

### 5.1. Harnessing NK Cells: Immune Checkpoint Inhibitors (ICIs)

To challenge infections, immune innate cells exploit their effector by a variety of activating receptors (aRs) including NCRs, NKG2D, and DNAM-1. These aRs sense pathogen-associated or endogenous molecules that are up-regulated or are expressed de novo at the cell surface under pathological conditions such as infections and/or tumors. Both events result in cytolytic activity and/or production of effector cytokines. Since during inflammation immune-mediated responses might also exert damage to self-tissues, sophisticated control mechanisms to avoid unwanted responses are needed.

These functions are mediated by a number of inhibitory receptors (iRs) called inhibitory ICs, which limit the threshold for effector cell activation and control homeostasis, resolution of inflammation, and self-tolerance. Tumors hijack inhibitory ICs to escape immune eradication. ICIs therapy was confirmed a powerful approach to cancer immunotherapy. In particular, antibodies (Abs) able to abrogate PD-1/PD-L1 interaction have demonstrated extraordinary activity in several types of cancers including metastatic melanoma and NSCLC. Monoclonal Abs (mAbs) targeting PD axis are able to prompt an effective antitumor response mainly through reinvigoration of exhausted PD-1^+^ CD8^+^ effectors T cells at the tumor site. However, resistance to PD-1/PD-L1 axis blockade remains a challenge for many patients. Besides specific T lymphocytes, also NK cells play an important role in anti-tumor immunity. Indeed, NK cells are potentially able to recognize and eliminate tumors that elude CD8+ T cell-mediated control by reducing HLA class I expression on their surface. Along this line, in long survey subjects, a lower degree of NK cytolytic potential has been correlated with cancer incidence [44]. Furthermore, several studies have provided evidence that in various solid malignancies the presence of TA-NK cells is associated with a better patient outcome [45].

The function of human NK cells is primarily regulated by classical ICs (including KIRs and NKG2A) specific for HLA-class I molecules which counteract the function of aRs. The fully anti-tumor potential of NK cells could be hindered by other non-classical ICs, recognizing ligands other than HLA class I molecules (PD-1, TIGIT, CD96, TIM-3, and LAG-3) (*see above section*). Of note, in hematological malignancies, a high expression of ligands for IC correlates with poor patients’ prognosis [46]. In TA-NK cells, the expression of some of these ICs can be modulated by signals present in the TME, thus inhibiting NK cell functions.

Thus, since their blocking may restore NK-cell responses against tumor cells, all these ICs may represent therapeutic targets. Due to the impressive inhibitory effect exerted by human mAbs blocking classical ICs on NK cells, they were the first to enter clinical phase. These include lirilumab (1-7F9, IPH2101) targeting KIR2DL1, KIR2DL2, and KIR2DL3 in patients with AML, myeloma, or solid tumors in AML [47,48] and IPH4102, a first-in-class monoclonal antibody targeting KIR3DL2 in patients with cutaneous T-cell lymphoma, predominantly those with Sézary syndrome. Unfortunately, only IPH4102 has shown, so far, promising clinical activity [49]. Dual ICIs therapy with nivolumab plus lirinumab in patients with recurrent resectable squamous cell carcinoma of the head and neck (SCCHN) is also being evaluated (NCT03341936).

An important improvement supporting the use in the clinic of mAbs recognizing NK cell-“specific” ICs has been obtained by Andrè and collaborators who analyzed the potential of an NKG2A blocking mAb (used either alone or combined other therapeutic mAbs) to unleash NK cell effector functions against HLA-E^+^ tumor cells [50].

Notably, the non-classical HLA class I molecule HLA-E is expressed by various human malignancies (such as lung, colon, pancreas, stomach, head and neck, and liver tumors), and NKG2A^+^ NK cells can be found in the tumor nest.

Monalizumab is an anti-NKG2A blocking mAb that not only boosts NKG2A^+^NK cell responses against HLA-E^+^ tumor cells but also promotes the effectiveness of durvalumab (a mAb that blocks PD-L1) by increasing the functional activity of NKG2A^+^PD-1^+^ NK cells against HLA-E^+^PD-L1^+^ target cells. Currently, clinical trials testing monalizumab either alone or combined with other mAbs (including anti-EGFR or anti-PD-L1) are active (*see next section*).

Several studies in humans have shown that NK cells from cancer patients express PD-1, which correlates with a lower anti-tumor activity [23,25,51,52]. Recently, it has been described that the therapeutic effect of PD-1 and PD-L1 blockade may rely also on the antitumor activity of NK cells [53]. Using several cancer mice models, Hsu and coworkers found that activated NK cells express PD-1 and that PD-1 engagement by PD-L1^+^ tumor cells potently suppress NK cell–mediated immunity to tumors [53]. Thus, the blockade of PD-1 or PD-L1, is able to activate an NK response that could be crucial for the full effect of PD-1/PD-L1 blockade. Our recently published results [54] correlate higher absolute numbers of circulating NK cells with longer overall survival (OS) in NSCLC patients treated with Nivolumab. These data are in line with what is already documented by [55], thus suggesting that the exact impact of NK cells on the response to nivolumab is an aspect that needs further studies.

TIGIT blockade can reverse the exhausted status of TA-NK cells [56], thus representing a potential new strategy to explore in immunotherapy. Since TIGIT could act as a negative regulator of NK cell functions, it represents an ideal molecule that can be targeted in checkpoint blockade strategies to boost NK tumor-immunity against CD155/PVR expressing cancers. Indeed, TIGIT has recently entered the spotlight as a promising IC target in cancer immunotherapy [57,58]. It is worth noting that IL-15 increased both DNAM-1/CD226 and TIGIT expression by TA-NK cells, thus in the presence of TIGIT-blocking mAbs, IL-15-activated NK cells can be triggered via DNAM-1/CD226. These alterations in both activating (CD226, the good one) and inhibitory (TIGIT, the bad one) receptors levels, together with TIGIT -targeted therapy may tip the balance in the net activating signaling output. It should also be stressed that translating TIGIT blockade into the clinic would be safer than the PD-1 or CTLA-4 blockade. Indeed, mice deficient for TIGIT do not show any sign of spontaneous autoimmunity or any defects in hematopoiesis. Thus far, the therapeutic role of mAbs blocking TIGIT (utilized alone or combined with anti-PD-1/PD-L1 mAbs) is being investigated in trials of patients with metastatic solid tumors (*see below section*). It has been described that in lung carcinoma [59] and gastric cancer patients [60] TIM-3 upregulation on peripheral blood NK cells is associated with reduced OS and advanced tumor stage, respectively. In patients suffering from esophageal tumors, NK cells expressing TIM-3 display an exhausted phenotype. Along this line, in these patients, high levels of TIM-3 on NK cells infiltrating the tumor nest are associated with tumor progression [61]. This clinical evidence suggests that in metastatic patients TIM-3 can act as a marker of exhaustion in NK cells, thus supporting the role of TIM-3 blocking mAbs in reinvigorating anti-tumor immunity. Further supporting these data, it has been described that in melanoma patients TIM-3-targeted therapy was able to restore NK cell function [62]. On the other hand, some studies described that TIM-3 can act as a stimulatory molecule able to promote T cell activation and differentiation [63,64]. Thus, since TIM-3 can also display a triggering function, in clinical use, anti-TIM- 3 blocking mAbs should be employed with care.

Lastly, whereas in mouse models it has been demonstrated that LAG-3 is able to impair NK cell activity against metastases [65], limited information exists on human LAG-3^+^ NK cells. In different tumor cell types, clinical trials investigating Relatlimab (a LAG-3 blocking mAb), used either alone or combined with other IC blocking mAbs, are active (*see below section*) (Figure 1).

### 5.2. Clinical Data on Therapeutic Approaches in Solid Tumors Involving Both Classical and Emerging/Non-Classical Immune Checkpoints

Monalizumab is an ICI active on NKG2A and thus able to activate anti-tumor activity of NK cells. This agent was evaluated in combination with cetuximab and durvalumab in a non-randomized, single-arm phase II trial involving treatment-naïve patients with recurrent or metastatic SCCHN (NCT02643550); in this study, the combination including monalizumab was characterized by acceptable safety profile and promising antitumor activity [66]. With regards to NSCLC, the activity of monalizumab was recently explored in the COAST trial (NCT03822351), an open-label phase II, randomized study in which patients who had been treated with chemo-radiation for inoperable stage III NSCLC were randomized to receive maintenance with durvalumab alone, the current standard of care, or durvalumab in combination with either oleclumab (an anti-CD73 agent) or monalizumab. With regards to the arm containing durvalumab plus monalizumab, the objective response rate (ORR) was 35.5%, while the median progression-free survival (mPFS) was 15.1 months, while ORR and mPFS for durvalumab alone were 17.9% and 6.3 months, respectively; additionally, the combination was generally well tolerated [67]. Currently, other clinical trials are exploring the role of anti-NKG2A treatment. Among these, the ongoing Precision Immuno-Oncology for Advanced Non-small Cell Lung Cancer Patients With PD-1 ICI Resistance (PIONeeR) trial is extremely promising, as it aims to enroll patients with NSCLC who had previously received ICIs and who will be treated with a combination of durvalumab and other agents, including monalizumab, in order to overcome acquired resistance to single-agent ICIs (NCT03833440) [https://clinicaltrials.gov/ct2/show/NCT03833440 (accessed on 22 September 2022)].

Lirilumab is another agent active on NK cells, as it masks KIR2D receptors, hence enhancing cytotoxicity of NK cells. After showing a manageable safety profile in dose-finding, phase I studies, both alone and in combination with an anti-PD-1 agent [47,68], the activity of lirilumab in combination with nivolumab in the neoadjuvant and adjuvant setting for squamous cell cancer of head and neck was explored in a single-arm phase II trial published by Hannah et al. In this study, the combination was generally well tolerated and achieved good outcomes in terms of 1-year disease-free survival (DFS) and 1-year OS, respectively, 55.2% and 85.7% [69]. Furthermore, lirilumab was employed in combination with epacadostat (an Indoleamine 2,3-dioxygenase 1, IDO-1, inhibitor) and nivolumab in the ECHO-208 trial; recently, the enrollment for this study was stopped, but its results are still not available (NCT 03347123) [https://clinicaltrials.gov/ct2/show/results/NCT03347123 (accessed on 22 September 2022)]. The relevant clinical data of these agents are summarized in Table 1.

Another novel drug class with potential activity on NK is represented by the inhibitors of TIGIT; indeed, TIGIT blockade is associated with prevention of NK exhaustion and enhancement of NK anti-tumor immunity [56]. Tiragolumab is a novel immune checkpoint inhibitor designed to target TIGIT, with promising activity in solid malignancies and especially in NSCLC. Indeed, in the phase II trial CITYSCAPE, 135 patients affected by advanced NSCLC with PD-L1 expression ≥1% were randomized (1:1) to receive atezolizumab (anti-PD-L1) plus tiragolumab or atezolizumab alone. At the interim analysis, the combination regimen achieved a statistically significant advantage in terms of PFS and numerical, albeit non-statistically significant, advantage in terms of OS over single-agent PD-L1 inhibitor; notably, in the sub-group analysis, the advantage achieved by the combination was pronounced among patients whose tumor harbored high PD-L1 expression (≥50%), but not among patients with PD-L1 between 1–49%. To date, the median OS has not been reported yet [70]. Currently, the role of tiragolumab is being explored in other settings for the management of NSCLC, such as maintenance after chemo-radiation for locally advanced disease (NCT04513). Another promising immune-related molecule is LAG-3, which has been firstly found on activated NK cells. Knockout LAG-3 mice have a decreased natural killer activity. Notably, it has been observed that LAG-3 plays a critical role in NKT cell function, and its expression results in decreased proliferation and functions of NKT cells, i.e., a cell subset that expresses both NK receptors and T cell receptors [71]. Relatlimab (anti-LAG-3) was employed in combination with nivolumab in a population of patients with advanced, pre-treated melanoma, resulting substantially safe and active [72]. Based on these results, the combination of relatlimab plus nivolumab was evaluated in a placebo-controlled, randomized phase II/III trial (RELATIVITY-047) designed to include patients with advanced melanoma. In this study, mPFS, the primary endpoint, was significantly improved in the experimental arm (relatlimab plus nivolumab) compared to placebo-nivolumab; hence, the combination of LAG-3 and PD-1 inhibition seems promising from a clinical perspective [73].

Notably, a co-formulation including nivolumab/relatlimab was approved for the use of metastatic melanoma by the American Food and Drug Administration (FDA) in March 2022 [https://www.fda.gov/drugs/resources–information–approved–drugs/fda–approves–opdualag–unresectable–or–metastatic–melanoma (accessed on 22 September 2022)] and by the European Medicine Agency (EMA) in July 2022 [https://www.ema.europa.eu/en/medicines/human/EPAR/opdualag (accessed on 22 September 2022)].

Finally, another molecule TIM-3 represents a promising target as its expression is down-regulated in activated NK cells [74]. The safety and activity of sabatolimab, an anti-TIM-3 antibody, administered alone or in combination with spartalizumab (an anti-PD-1 antibody) were explored in a phase I/II trial designed to enroll patients with solid tumors; the safety profile of the combination was generally manageable at the recommended phase II dose identified in the dose escalation; indeed, the maximum tolerated dose was not reached. With regards to activity, initial responses were observed, thus leading to further clinical development of this combination [75]. The relevant clinical data of the aforementioned agents are summarized in Table 2.

In addition to the agents specifically designed to act on IC expressed by NK cells, there are other relevant drugs, active on novel, emerging immune checkpoints; such checkpoints are acknowledged to influence, among other immune cells, also NK. Elotuzumab, a mAb directed on signaling lymphocytic activation molecule F7 (SLAM7), has shown the ability to induce NK cell infiltration and cytotoxicity, albeit this activity was specifically observed in multiple myeloma pre-clinical models [80]. In a single-arm, phase II trial, patients with multiple myeloma received elotuzumab plus pomalidomide, carfilzomib, and low-dose dexamethasone, with good tolerability and promising activity [76]. To date, knowledge on the potential role of elotuzumab in solid tumors is still limited.

Among the molecule of interest for immunotherapeutic agents, IDO-1 has emerged as a potential novel target for immune checkpoint blockade (ICB), as it is known to inhibit proliferation and activity of cytotoxic T cells and NK cells [81]. The IDO1 inhibitor epacadostat has been evaluated in several trials designed to assess its tolerability and activity. In the phase I ECHO-110 trial, epacadostat in combination with atezolizumab showed a globally manageable safety profile, as well as some level of antineoplastic activity, thus leading to further trial development [77]. Similarly, in a phase I/II trial, ECHO-202/KEYNOTE-037, epacadostat in combination with pembrolizumab was generally manageable and active, especially among patients with melanoma and NSCLC [78]. Based on these results, the efficacy of epacadostat plus pembrolizumab was evaluated in a phase III, placebo-controlled trial involving 706 patients affected by advanced melanoma and naïve from ICIs; notably, the trial did not meet any of its co-primary endpoints, PFS and OS, and the author concluded that the clinical role of IDO-1 inhibitors remains uncertain [79].

## 6. Role of NK Cells in Creating a More Inflamed Environment (to Prepare the Ground for ICI)

Several studies have implicated an important role of NK cells in tumor immune surveillance. Many results were derived from mouse models, which were either depleted of NK cells or impaired in conventional NK cell activities. Remarkably, these studies also demonstrated an exceptional capacity of NK cells to resist the hematogenous spread of experimental and spontaneous tumor metastases [82]. These preclinical data were further supported by observational studies in humans, even evaluating large cohorts, where NK cell deficiencies [83], as well as lower NK cell activity in peripheral blood [44], could be associated to a higher risk of developing various types of cancer. Regarding lung cancers, the prognostic value of infiltrating NK cells in resected tumors still needs to be defined. Primarily, this is due to the limited number of studies performed and the small size of cohorts analyzed in each study. Moreover, it should be noted that these analyses were performed using markers (i.e., CD57 and/or CD56) not exclusive of NK cells, but potentially expressed by other immune/non-immune cell types. As such, some initial studies indicated that the presence of NK cells in the immune infiltrate was associated with a lower risk of relapse and/or longer survival [84,85] while subsequent studies, performed using the more specific marker NKp46, failed to find an impact of high number of intra-tumoral NK cells (at early stages of disease) on OS [86]. However, recent reports showed that the number of infiltrating CD56^+^CD16^+^ NK cells in lung cancer tissue positively correlated to patient survival [87]. Moreover, an “immune cluster” with a signature of NK cells and/or plasma cells was discovered in a limited number of the analyzed NSCLC cases (5%) and was associated with improved survival. Remarkably, this subgroup showed a favorable prognosis despite the lack of markers for T cells or T-cell activation [88].

The low number of infiltrating NK cells has raised questions about the actual function of this cell population at the tumor site. However, even if generally underrepresented in solid tumors, these cells could contribute to anti-tumor immune responses by means of their cytolytic activity or their remarkable capacity to produce cytokines and chemokines that recruit and activate (or potentially suppress) other hematopoietic cells [89]. Several reports have found that cytotoxic CD56^dim^ Perforin^high^ NK cells are quite excluded from tumor tissue. Conversely, CD56^bright^ Perforin^low^ NK cells represent the main NK cell population infiltrating human cancer tissues, and at least for some tumor types, such as NSCLC and breast cancer, the ratio of cytotoxic CD56^dim^ Perforin^high^ to non-cytotoxic CD56^bright^ Perforin^low^ is completely inverted when compared to the matched normal tissues. Interestingly, the relative accumulation of CD56^bright^ Perforin^low^ was associated with a switch in chemokine expression patterns of tissues upon the neoplastic transformation [90]. Thus far, this phenomenon remains an interesting and poorly explored aspect in the field. Whether this may represent a specific strategy used by the immune system to control tumor growth or rather a mechanism of tumor immune evasion is yet to be determined. As such, the presence of non-cytotoxic CD56^bright^ Perforin^low^ NK cells, which are devoid of CD16 and represent the dominant NK cell subset infiltrating several human solid cancers, could limit the response of agents aimed at boosting NK-mediated ADCC. Overall, defining how NK cell functional diversity integrates into innate and adaptive immune responses to cancer represents a critical challenge. DCs, the most efficient APCs of the immune system, are now established as a critical immune effector based on their ability to induce anti-tumor T cell immunity and response to immunotherapies. Among different tumor-infiltrating DC phenotypes found across solid human cancers, conventional type 1 DCs (cDC1s) are specialized in antigen cross-presentation and CD8+ T cell activation. Accordingly, in human tumors, gene expression signatures related to cDC1s have been correlated with better clinical prognosis and response to ICB [91,92]. Therefore, the possibility of recruiting cDC1 into tumors, as well as improving their functionality, could prove to be useful strategies for increasing antitumor immunity and response to immunotherapies. Remarkably, an additional role of NK cells in the immune response to cancer has been demonstrated by recent publications that showed NK cells controlling the levels of intratumoral cross-presenting cDC1s, by expression of FLT3 Ligand [92] and chemokines, such as CCL5 (RANTES) and XCL-1/XCL-2 [91,93]. In patients with melanoma, levels of NK cells and intratumoral cDC1s even positively correlated with increased survival and predicted response to anti-PD-1 therapy [92]. In this respect, NK cells have been suggested as a “spark” that ignites immune cell infiltration and inflammation in the tumor [94]. It is noteworthy that unique clusters of NK cells characterized by high expression of XCL1/2 transcripts were also identified among total NK cells isolated from melanoma metastasis [95]. DCs and NK cells can reciprocally engage in a bi-directional activation that can influence the outcome of adaptive immunity, by influencing the development of T helper-1 (T_H_-1) cells and cytotoxic T lymphocytes (CTLs), both essential for an effective anti-tumor immune response. Granulocyte-macrophage-colony stimulating factor (GM-CSF) is a potent cytokine promoting the differentiation of myeloid cells and is essential for the differentiation of dendritic cells, which are responsible for processing and presenting tumor antigens for the priming of CTLs [96]. Interestingly, NK cells may potentially be a major source of Granulocyte-Macrophage Colony-Stimulating Factor (GM–CSF) in tumors, especially CD56^bright^ NK cells, which are enriched in neoplastic tissues [90] and represent the NK cell subset producing the higher levels of this cytokine [4].

Overall, recruitment and modulation of APCs at the tumor site could have a great impact on cancer immune surveillance, given the positive association of CD8^+^ T cell infiltration with longer survival in NSCLC patients [97,98]. Interestingly, in a mouse model of lung adenocarcinoma, stimulation of tumor-infiltrating NK cells by a conditional expression of activating NK cell ligands led to an increase of tumor-specific T cells. Mechanistically, the accumulation of adaptive immune cells was not due to overt signs of cytotoxicity in tumors against tumor cells but, rather, to the direct production of chemokines, such as CCL5, or indirectly, to the stimulation of APCs, as suggested by the authors [99]. Finally, Zemek RM et al. demonstrated, in both an animal tumor model of mesothelioma and datasets from patients, that the presence of activated NK cells in the TME and expression of immune response-related genes characterized by Signal transducer and activator of transcription 1 (STAT1) activation can correlate with the clinical response to ICI [100].

## 7. Conclusions

Although ICI immunotherapy is well positioned as a safe anti-tumor therapy, important questions remain open. Elucidating the key parameters that unleash not only the activity and reinvigoration of T cells but also NK cell potential will be important as the field progresses into developing approaches to address challenges specific to each different neoplastic disease indication.

The relief from NK-specific IC implies potential therapeutic advantages related to the quite modest autoimmune burden and significant anti-tumor activity of NK cells. However, for the emerging NK cell therapy programs, decisions depend on still open issues, such as the altered responsiveness of NK cells in the patients and the limited persistency of NK cell activation in vivo or in adoptively transferred NK cells. The generation of engineered molecules, combining arms specific for different NK receptors and targeting tumor epitopes, represents an important element for the assembly of new therapeutic strategies. These multivalent molecules (bi- tri- or tetra-specific engagers) [101,102,103,104] may combine multiple therapeutic effects, depending on the assembled specificities, stimulating triggering NK receptors and cytokine receptors, blocking ICs, and targeting tumor cells. Another important approach deals with the use of combined cytokine cocktails (IL-12, IL-18, IL-15) to get the so-called Cytokine-Induced Memory-Like (CIML) NK cells. These cells are able to “remember” the initial cytokine boost and maintain their increased responsiveness and even persist in the patients after adoptive transfer [105]. Finally, engineered NK cell products expressing chimeric Antigen Receptors (NK-CAR) can be obtained from fresh NK cells or from precursor cells (including induced-Pluripotent Stem Cells—iPSC) [106].

Most of these tools, including the ICI, are being studied in clinics and hold promise, but the real frontier in the field is the search for appropriate therapeutic combination to maximize the anti-tumor power of NK cells in the different specific pathologic conditions [107].

## Figures and Tables

**Figure 1 cancers-14-05046-f001:**
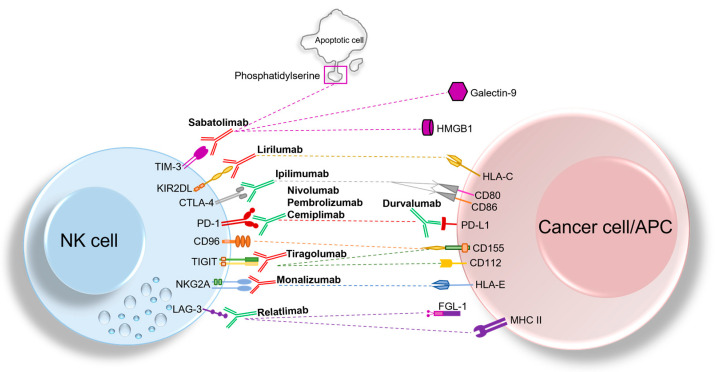
Overview of classical and non-classical ICs expressed by NK cells and their cognate ligands expressed either on cancer cells or antigen-presenting cells (APCs). The blocking mAbs that are currently being investigated in clinical trials, including anti- TIM-3, anti-NKG2A, anti-KIR, and anti-TIGIT mAbs are depicted in red, while blocking mAbs that are approved by FDA and currently used in clinic (such as anti-PD-1, anti-CTLA-4, and anti-LAG-3 mAbs) are represented in green. Blockade of ICs could recover NK cell anti-tumor activity, thereby representing a promising approach for immunotherapy.

**Table 1 cancers-14-05046-t001:** Relevant clinical data of immune checkpoint inhibitors specifically designed to enhance anti-neoplastic NK cell activity. Legend -> DCR: disease control rate; DLT: dose-limiting toxicity; DFS: disease-free survival; NSCLC: non-small cell lung cancer; SCCHN: squamous cell carcinoma of head and neck; OS: overall survival; PFS: progression-free survival; TRAE: treatment-related adverse event. * in the COAST trial, durvalumab was employed alone or in combination with oleclumab or monalizumab. In this table, we are considering the outcomes of the regimen containing durvalumab and monalizumab.

Target	Agent	Study	Population	Regimens	Outcomes
NKG2A	Monalizumab	Colevas et al. (single-arm phase II trial) [66]	40 patients with recurrent/advanced SCCHN	Durvalumab plus cetuximab plus monalizumab	ORR: 32.5% Median PFS: 6.9 months; 12 month-OS: 59%
		COAST (open-label, randomized, phase II trial) [67]	189 patients with stage III NSCLC candidates for maintenance after chemo-radiation	Durvalumab plus monalizumab or durvalumab plus oleclumab vs. durvalumab alone *	Monalizumab + durvalumab arm vs. durvalumab alone Confirmed ORR: 35.5% vs. 17.9% Median PFS: 15.1 vs. 6.3 months
KIR 2D	Lirilumab	Vey et al. (phase I trial) [47]	37 patients with solid or hematologic malignancies	Escalating doses of lirilumab	No DLT were identified; full KIR occupancy (>95%) was achieved with all dosages
		Armand et al. (phase I trial) [68]	72 patients with classical Hodgkin lymphoma, non-Hodgkin lymphoma, or multiple myeloma	Lirilumab plus nivolumab	ORR: 76% Grade 3–4 TRAE: 15%
		Hannah et al. (single-arm phase II trial) [69]	28 patients with operable SCCHN	Lirilumab plus nivolumab 7–21 days before surgery, followed by 6 cycles of adjuvant lirilumab plus nivolumab	DCR at surgery: 96% 1-year DFS: 55.2% 1-year OS: 85.7% Grade 3+ TRAE: 11%

**Table 2 cancers-14-05046-t002:** Relevant clinical data of immunotherapy agents with potential effect on NK cells. Legend -> AE: adverse event; CI: confidence interval; CBR: clinical benefit rate; HR: hazard ratio; ITT: intent-to-treat; MTD: maximum-tolerated dose; NR: not reached; NSCLC: non-small cell lung cancer; ORR: objective response rate; RCC: renal cell carcinoma; SAE: severe adverse events. * *p* values were not reported in the interim analysis.

Target	Agent	Study	Population	Regimens	Outcomes
SLAM7	Elotuzumab	Yashar et al. (single-arm phase II trial) [76]	13 patients with high-risk relapsed/refractory multiple myeloma	Elotuzumab plus pomalidomide, carfilzomib, and low-dose dexamethasone	ORR: 45.4%; CBR: 54.5% SAE rate: 31%
TIGIT	Tiragolumab	CITYSCAPE (open-label, randomized, phase II trial) [70]	135 patients with advanced NSCLC (PD-L1 ≥ 1%)	Tiragolumab plus atezolizumab vs. atezolizumab alone	**INTERIM ANALYSIS *** ITT population Median PFS: 5.6 vs. 3.9 months; HR: 0.62 (95% CI: 0.42–0.91) Median OS: 23.2 vs. 14.5 months; HR: 0.69 (95% CI: 0.44–1.07) PD-L1 ≥ 50% Median PFS: 16.6 vs. 4.1 months; HR: 0.29 (95% CI: 0.15–0.53) Median OS: NR vs. 12.8 months; HR: 0.23 (95% CI: 0.10–0.53) PD-L1 between 1–49% Median PFS: 4.0 vs. 3.6 months; HR: 1.07 (95% CI: 0.67–1.71) Median OS: 13.3 vs. 14.5 months; HR: 1.16 (95% CI: 0.70–1.94)
IDO1	Epacadostat	ECHO-110 (phase Ib trial) [77]	29 patients with advanced, pre-treated NSCLC	Epacadostat (increasing doses) plus atezolizumab	Grade ≥ 3 AEs rate: 24% 8 patients achieved stable disease 1 patient achieved partial response
		ECHO-202/KEYNOTE-037 (phase I/II trial) [78]	62 patients with advanced solid tumors	Epacadostat (increasing doses) plus pembrolizumab	Grade ≥ 3 AEs rate: 24% ORR melanoma: 12/22 patients (55%) ORR NSCLC: 5/12 patients (42%) ORR RCC: 2/11 patients (18%)
		ECHO-301/KEYNOTE-252 (placebo-controlled, randomized, phase III trial) [79]	706 patients with unresectable stage III or IV melanoma previously untreated with PD-1 or PD-L1 checkpoint inhibitors	Epacadostat plus pembrolizumab vs. placebo plus pembrolizumab	Median PFS: 4.7 vs. 4.9 months; HR: 1.00 (95% CI: 0.83–1.21; *p* = 0.52) Median OS: not reached in any arm; HR: 1.13 (95% CI: 0.86–1.49; *p* = 0.81)
LAG-3	Relatlimab	Ascierto et al. (phase I/IIa trial) [72]	43 patients pre-treated with ICIs for melanoma	Relatlimab plus nivolumab	ORR: 16%; DCR: 45% Any grade AEs rate: 46%; Grade 3–4 AEs rate: 9%
		RELATIVITY-047 (placebo-controlled, randomized, phase II/III trial) [73]	714 patients with previously untreated advanced melanoma	Relatlimab plus nivolumab vs. placebo plus nivolumab	Median PFS: 10.1 vs. 4.6 months; HR: 0.75 (95% CI: 0.62–0.92; *p* = 0.006) Grade 3–4 AEs rate: 18.9% vs. 9.7%
TIM-3	Sabatolimab	Curigliano et al. (phase I/II trial) [75]	219 patients with solid tumors	Escalating doses of sabatolimab alone or sabatolimab plus spartalizumab	MTD not reached No response with sabatolimab alone ORR (sabatolimab plus spartalizumab): 6%

## Data Availability

Not applicable.

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
