# Peer review of "Immune Checkpoint Blockade: A Strategy to Unleash the Potential of Natural Killer Cells in the Anti-Cancer Therapy"

_cancers, 2022, doi:10.3390/cancers14205046_

Round 1
Reviewer 1 Report
This review article provides an overview about the exploration of NK cells in immune checkpoint blockage therapy for cancer treatment. Authors described the significance of NK cells, their associated mechanism in immune checkpoint inhibitors in very well manner, more specifically in terms of clinical efficacy.
Before publication the manuscript requires some suggestions to improve the overall concept:
1. Introduction part must be improved in terms of incorporation of more content related to immunotherapy and immune checkpoint. Authors can refer to the latest article in this field:
https://doi.org/10.3390/ph15030335
https://doi.org/10.2174/1871520620666200728121929
2. Authors must include a figure to represent the section 5/6.
3. Authors have described the therapeutics used for the anticancer therapy in both the sections (section 5 and 6), if possible try to present under a common heading/subheading.
4. Typographical errors are present at several places throughout the manuscript. Authors must proofread the whole manuscript for possible errors. For example,
Line number: 118 …….”in vivo [19]”
Line number 165 ….. [33] [34],
Line number 202……. tumor progression[47]
Line number 260…… (NCT04513……..
5. Authors must used full form for abbreviation wherever used first time.
6. Conclusion section should be improved in terms of future perspectives.
Author Response
First of all we want to thank the review since her/his advices have been found very useful in order to improve our manuscript. All the modifications made to the paper in order to follow the suggestions are written in red.
Point by point reply:
- Introduction part must be improved in terms of incorporation of more content related to immunotherapy and immune checkpoint. Authors can refer to the latest article in this field: R: As suggested by the assessor we have improved the introduction part and the latest articles in the field, have been quoted.
- Authors must include a figure to represent the section 5/6. R: As suggested, we have included a figure to represent section 5.
- Authors have described the therapeutics used for the anticancer therapy in both the sections (section 5 and 6), if possible try to present under a common heading/subheading. R: Both sections are now under a common heading.
- Typographical errors are present at several places throughout the manuscript. Authors must proofread the whole manuscript for possible errors. R: We have carefully proofread the manuscript for typo errors.
- Authors must use full form for abbreviation wherever used first time. R: We have used the full form name when used for the first time
- Conclusion section should be improved in terms of future perspectives. R: The conclusion section has been implemented as suggested
Reviewer 2 Report
The review by Melania Grottoli et al. titled “Immune checkpoint blockade: a strategy to unleash the potential of Natural Killer cells in the anti-cancer therapy.” provides an overview of the recent findings concerning the role of classical and non-classical immune checkpoint molecules and receptors that regulate natural killer cell function. The review is well-written and pleasant reading. It focuses on the rationale, mechanisms of action, and clinical efficacy of the NK cell-based checkpoint blockade therapy. The topic is certainly timely, with many important discoveries on this topic made in the past few years. Also, the topic should interest a large number of scientists interested in the efficacy of ICB. The authors reviewed the recent literature well and summarized the key findings. Despite a couple of minor points (see below), this reviewer support publication of the manuscript once those few points are solved.:
Some punctuation marks need to be checked. Line 217, 268, 355. Full stops are missing.
Line 343. This reference “Barry K C Nat Med 2018 PMID: 29942093” is the same that 77
Author Response
First of all, we want to thank the review for her/his kindness and comments.
As recommended by the assessor we have carefully proofread the manuscript for typo errors.
Reviewer 3 Report
The work is great, but, there are so many abbreviations. I would recommend the authors to either provide full forms of the abbreviations initially at their first mention or to include a table with all the abbreviations and their respective full forms.
There are many strong statements that were made that had either 1 or no citations whatsoever.

Author Response
First of all we want to thank the review since her/his advices have been found very useful in order to improve our manuscript. The modifications made to the paper in order to follow the suggestions are written in green.
We have provided in the text the full forms of the abbreviations at their first mention. The have revised the text following the suggestions found in the attached PDF file (peer-review-22774097.v1.pdf)
Round 2
Reviewer 3 Report
all good. No further revisions